# AVOIDING SPURIOUS CORRELATIONS VIA LOGIT CORRECTION

**Sheng Liu**[1][*] **Xu Zhang**[2] **Nitesh Sekhar**[2] **Yue Wu**[2]
**Prateek Singhal**[2] **Carlos Fernandez-Granda**[1]
[1]New York University, USA; `shengliu@nyu.edu` [2]Amazon Alexa AI, USA

## ABSTRACT

Empirical studies suggest that machine learning models trained with empirical risk minimization (ERM) often rely on attributes that may be spuriously correlated with the class labels. Such models typically lead to poor performance during inference for data lacking such correlations. In this work, we explicitly consider a situation where potential spurious correlations are present in the majority of training data. In contrast with existing approaches, which use the ERM model outputs to detect the samples without spurious correlations and either heuristically upweight or upsample those samples, we propose the logit correction (LC) loss, a simple yet effective improvement on the softmax cross-entropy loss, to correct the sample logit. We demonstrate that minimizing the LC loss is equivalent to maximizing the group-balanced accuracy, so the proposed LC could mitigate the negative impacts of spurious correlations. Our extensive experimental results further reveal that the proposed LC loss outperforms state-of-the-art solutions on multiple popular benchmarks by a large margin, an average 5.5% absolute improvement, without access to spurious attribute labels. LC is also competitive with oracle methods that make use of the attribute labels.[†]

## 1 INTRODUCTION

In practical applications such as self-driving cars, a robust machine learning model must be designed to comprehend its surroundings in rare conditions that may not have been well-represented in its training set. However, deep neural networks can be negatively affected by spurious correlations between observed features and class labels that hold for well-represented groups but not for rare groups. For example, when classifying stop signs versus other traffic signs in autonomous driving, 99% of the stop signs in the United States are red. A model trained with standard empirical risk minimization (ERM) may learn models with low average training error that rely on the spurious background attribute instead of the desired "STOP" text on the sign, resulting in high average accuracy but low worst-group accuracy (e.g., making errors on yellow color or faded stop signs). This demonstrates a fundamental issue: models trained on such datasets could be systematically biased due to spurious correlations presented in their training data (Ben-Tal et al., 2013; Rosenfeld et al., 2018; Beery et al., 2018; Zhang et al., 2019). Such biases must be mitigated in many fields, including algorithmic fairness (Du et al., 2021), machine learning in healthcare (Oakden-Rayner et al., 2020; Liu et al., 2020b; 2022a), and public policy Rodolfa et al. (2021).

Formally, spurious correlations occur when the target label is mistakenly associated with one or more confounding factors presented in the training data. The group of samples in which the spurious correlations occur is often called the *majority group* since spurious correlations are expected to occur in most samples, while the *minority groups* contain samples whose features are not spuriously correlated. The performance degradations of ERM on a dataset with spurious correlation (Nagarajan et al., 2021; Nguyen et al., 2021) are caused by two main reasons: 1) the geometric skew and 2) the statistical skew. For a robust classifier, the classification margin on the minority group should be much larger than that of the majority group (Nagarajan et al., 2021). However, a classifier trained with ERM maximizes margins and therefore leads to equal training margins for the majority and

---

[*]Work primarily done during an internship at Amazon.
[†]Code is available at `https://github.com/shengliu66/LC`.

minority groups. This results in geometric skew. The statistical skew is caused by slow convergence of gradient descent, which may cause the network to first learn the "easy-to-learn" spurious attributes instead of the true label information and rely on it until being trained for long enough (Nagarajan et al., 2021; Liu et al., 2020a; 2022b).

To determine whether samples are from the majority or minority groups, we need to know the group information during training, which is impractical. Therefore, many existing approaches consider the absence of group information and first detect the minority group (Nguyen et al., 2021; Liu et al., 2021b; Nam et al., 2020) and then upweight/upsample the samples in the minority group during training (Li & Vasconcelos, 2019; Nam et al., 2020; Lee et al., 2021; Liu et al., 2021a). While intuitive, upweighting only addresses the statistical skew (Nguyen et al., 2021), and it is often hard to define the weighted loss with an optimal upweighting scale in practice. Following Menon et al. (2013); Collell et al. (2016) on learning from imbalanced data, we argue that the goal of training a debiased model is to achieve a high average accuracy over all groups (Group-Balanced Accuracy, GBA, defined in Sec. 3), implying that the training loss should be Fisher consistent with GBA (Menon et al., 2013; Collell et al., 2016). In other words, the minimizer of the loss function should be the maximizer of GBA.

In this paper, we revisit the logit adjustment method (Menon et al., 2021) for long-tailed datasets, and propose a new loss called logit correction (LC) to reduce the impact of spurious correlations. We show that the proposed LC loss is able to mitigate both the statistical and the geometrical skews that cause performance degradation. More importantly, under mild conditions, its solution is Fisher consistent for maximizing GBA. In order to calculate the corrected logit, we study the spurious correlation and propose to use the outputs of the ERM model to estimate the group priors. To further reduce the geometrical skew, based on MixUp (Zhang et al., 2018), we propose a simple yet effective method called Group MixUp to synthesize samples from the existing ones and thus increase the number of samples in the minority groups.

The main contributions of our work include:

- We propose logit correction loss to mitigate spurious correlations during training. The loss ensures the Fisher consistency with GBA and alleviates statistical and geometric skews.
- We propose the Group MixUp method to increase the diversity of the minority group and further reduce the geometrical skew.
- The proposed method significantly improves GBA and the worst-group accuracy when the group information is unknown. With only 0.5% of the samples from the minority group, the proposed method improves the accuracy by 6.03% and 4.61% on the Colored MNIST dataset and Corrupted CIFAR-10 dataset, respectively, over the state-of-the-art.

## 2 RELATED WORK

Spurious correlations are ubiquitous in real-world datasets. A typical mitigating solution requires to first detect the minority groups and then design a learning algorithm to improve the group-balanced accuracy and/or the worst-group accuracy. We review existing approaches based on these two steps.

**Detecting Spurious Correlations.** Early researches often rely on the predefined spurious correlations (Kim et al., 2019; Sagawa et al., 2019; Li & Vasconcelos, 2019). While effective, annotating the spurious attribute for each training sample is very expensive and sometimes impractical. Solutions that do not require spurious attribute annotation have recently attracted a lot of attention. Many of the existing works (Sohoni et al., 2020; Nam et al., 2020; Liu et al., 2021a; Zhang et al., 2022) assume that the ERM model tend to focus on spurious attribute (but may still learn the core features Kirichenko et al. (2022); Wei et al. (2023)), thus "hard" examples (whose predicted labels conflict with the ground-truth label) are likely to be in the minority group. Sohoni et al. (2020); Seo et al. (2022), on the other hand, propose to estimate the unknown group information by clustering. Our work follows the path of using the ERM model.

**Mitigating Spurious Correlations.** Previous works (Nagarajan et al., 2021; Nguyen et al., 2021) show that the geometric skew and the statistical skew are the two main reasons hurting the performance of the conventional ERM model. Reweighting (resampling), which assigns higher weights (sampling rates) to minority samples, is commonly used to remove the statistical skew (Li & Vasconcelos, 2019;

Nam et al., 2020; Lee et al., 2021; Liu et al., 2021a). While intuitive, reweighting has limited effects to remove the geometric skew (Nagarajan et al., 2021). Also, there are surprisingly few discussions on how to set the optimal weights. We argue that the reweighting strategy should satisfy the Fisher consistency (Menon et al., 2013), which requires that the minimizer of the reweighted loss is the maximizer of the balanced-group accuracy (see Sec. 4.1). On the other hand, synthesizing minority samples/features is widely utilized in removing the geometric skew. Minderer et al. (2020); Kim et al. (2021) propose to directly synthesize minority samples using deep generative models. Yao et al. (2022); Han et al. (2022) synthesize samples by mixing samples across different domains. While synthesizing the raw image is intuitive, the computation complexity can be high. DFA (Lee et al., 2021) mitigates this issue by directly augmenting the minority samples in the feature space.

DFA (Lee et al., 2021) is the most related work to our approach. It applies reweighting to reduce the statistical skew and feature swapping to augment the minority feature, thus removing the geometric skew. However, our approach uses logit corrected loss and is proved to be Fisher consistent with the group-balanced accuracy. The proposed logit corrected loss has a firmer statistical grounding and can reduce both the statistical skew and the geometric skew. By combining the logit correction loss and the proposed Group MixUp, our approach outperforms DFA, especially on the dataset containing very few minority samples.

## 3    PROBLEM FORMULATION

Let's first consider a regular multi-class classification problem. Given a set of $n$ training input samples $\mathcal{X} = \{(\mathbf{x}_i, y_i)\}, i = 1, \ldots, n$, where, $\mathbf{x} \in \mathbb{R}^d$ has a input dimension of $d$ and $y \in \mathcal{Y} = \{1, \ldots, L\}$ with a total number of $L$ categories. Our goal is to learn a function (neural network), $f(\cdot) : \mathcal{X} \to \mathbb{R}^L$, to maximize the classification accuracy $P_{\mathbf{x}}(y = \arg\max_{y' \in \mathcal{Y}} f_{y'}(\mathbf{x}))$. With ERM, we typically minimize a surrogate loss, *e.g.*, softmax cross-entropy $\mathcal{L}_{CE}(\cdot)$ where,

$$\mathcal{L}_{CE}(y, f(\mathbf{x})) = \log\left(\sum_{y'} e^{f_{y'}(\mathbf{x})}\right) - f_y(\mathbf{x}). \tag{1}$$

We assume there is a spurious attribute $\mathcal{A}$ with $K$ different values in the dataset. Note that $K$ and the number of classes $L$ may not be equal. We define a combination of one label and one attribute value as a group $g = (a, y) \in \mathcal{A} \times \mathcal{Y}$. The spurious correlation means an attribute value $a$ and a label $y$ commonly appear at the same time. Different from ERM, the goal of training a model to avoid spurious correlation is to maximize the Group-Balanced Accuracy (GBA):

$$GBA(f) = \frac{1}{KL} \sum_{g \in \mathcal{G}} \mathbf{P}_{\mathbf{x}|(y,a)=g}\left(y = \arg\max_{y' \in \mathcal{Y}} f_{y'}(\mathbf{x})\right). \tag{2}$$

Note that spurious correlation is "harmful" when it is not present during the evaluation. The spurious attribute can not be disentangled from other features if spurious correlations are present in all samples.

## 4    OUR APPROACH: LOGIT CORRECTION

Following Nam et al. (2020), we adopt a two-branch network (as shown in Figure 1). The top branch (denoted as $\hat{f}(\cdot)$) is a network trained with ERM using generalized cross-entropy (GCE) loss Zhang & Sabuncu (2018):

$$\mathcal{L}_{GCE} = \frac{1 - \hat{p}(\mathbf{x})^q}{q}, \tag{3}$$

where $\hat{p}(\mathbf{x})$ represents the probability outputs (after a softmax layer) of the ERM model $\hat{f}$, $q \in [0, 1)$ is a hyperparameter. Compared to the standard cross-entropy loss, the gradient of GCE loss upweights examples where $\hat{p}(x)$ is large, which intentionally biases $\hat{f}$ to perform better on majority (easier) examples and poorly on minority (harder) examples. The second (bottom) branch is trained to learn from the first branch. To be more specific, we use the probability output of the first branch to correct the logit output of the second branch. We further adopt Group MixUp to increase the number of unique examples in minority groups. The details of the method are demonstrated in the following sections.

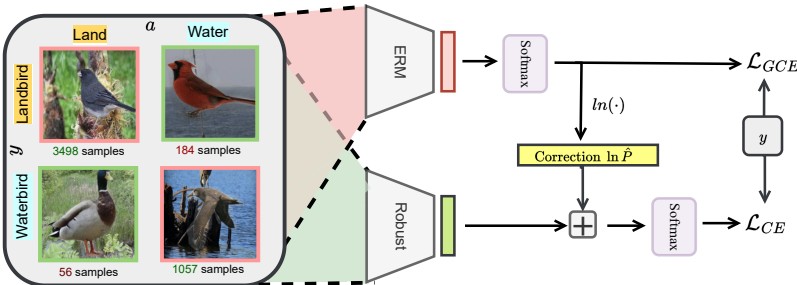

Figure 1: The overview of our proposed logit correction approach on the Waterbirds dataset, where the background (water/land) is spuriously correlated with the foreground (waterbird/landbird). Most training samples belong to the group where the background matches the bird type (highlighted in red); While only a small fraction belongs to the groups where the background mismatches the bird type (highlighted in green). To address this issue, we train both an ERM network and a robust network simultaneously. The ERM network is trained with a generalized cross-entropy (GCE) loss that intentionally biases it toward the majority groups. The logit correction loss corrects the logits of the robust network by a term $\hat{p}$, which is generated by the probability predictions of the ERM network. After logit correction, the robust network is trained with the standard cross-entropy loss.

### 4.1 LOGIT CORRECTION AS MAXIMIZING GBA

Recall that our goal is to maximize GBA in Eq 2, which depends on the (unknown) underlying distribution $\mathbf{P}(x, y, a)$, the Bayes-optimal prediction function under this setting is $f^* \in \arg\max_f GBA(f)$.

**Proposition 1.** *Let $P(y, a)$ be the prior of group $(y, a)$, and $P(y, a|\mathbf{x})$ is the true posterior probability of group $(y, a)$ given $\mathbf{x}$, the prediction:*

$$\arg\max_{y \in \mathcal{Y}} f_y^*(\mathbf{x}) = \arg\max_y \sum_a \frac{P(y, a|\mathbf{x})}{P(y, a)} = \arg\max_y \sum_a \frac{P(y|a, \mathbf{x})P(a|\mathbf{x})}{P(y, a)} \tag{4}$$

*is the solution to Eq. 2. See proof in Appendix A.*

Assume each example $\mathbf{x}$ can only take one spurious attribute value (e.g. waterbirds can either be on the water or land, and can not be on both), that is to say, the prior probability $P(a|\mathbf{x})$ is 1 when the spurious attribute $a = a_{\mathbf{x}}$ and 0 otherwise. We have

$$\arg\max_{y \in \mathcal{Y}} f_y^*(\mathbf{x}) = \arg\max_y P(y|a_{\mathbf{x}}, \mathbf{x})/P(y, a_{\mathbf{x}}). \tag{5}$$

Note that although $a_{\mathbf{x}}$ is unknown in the dataset, it can be estimated using the outputs of ERM model (see Sec. 4.2).

Because we are using the second branch to estimate the posterior probability $P(y|a_{\mathbf{x}}, \mathbf{x})$, supposing the underlying class probability $P(y|a_{\mathbf{x}}, \mathbf{x}) \propto \exp(f(\mathbf{x}))$ for an (unknown) scorer $f$, we can rewrite Eq. 5 as

$$\arg\max_{y \in \mathcal{Y}} f_y^*(\mathbf{x}) = \arg\max_{y \in \mathcal{Y}} \exp(f_y(\mathbf{x}))/P(y, a_{\mathbf{x}}),$$
$$= \arg\max_{y \in \mathcal{Y}} (f_y(\mathbf{x}) - \ln P(y, a_{\mathbf{x}})). \tag{6}$$

In other words, optimizing the GBA solution $f^*$ is equivalent to optimizing the ERM solution $f$ minus the logarithm of the group prior $\ln P(y, a_{\mathbf{x}})$. In practice, we could directly bake logit correction into the softmax cross-entropy by compensating the offset. Specifically, we can use the corrected logits $f(\mathbf{x}) + \ln \hat{P}_{y, a_{\mathbf{x}}}$ instead of the original logits $f(\mathbf{x})$ to optimize the network, where $\hat{P}_{y, a_{\mathbf{x}}}$ are estimates of the group priors $P(y, a_{\mathbf{x}})$. Such that by optimizing the loss function as usual, we can derive the solution for $f^*$ (Menon et al., 2021). The logits corrected softmax cross entropy function can be written as,

$$\mathcal{L}_{LC}(y, f(\mathbf{x})) = \log \left( \sum_{y'} e^{f_{y'}(\mathbf{x}) + \ln \hat{P}_{y', a_{\mathbf{x}}}} \right) - \left( f_y(\mathbf{x}) + \ln \hat{P}_{y, a_{\mathbf{x}}} \right). \tag{7}$$

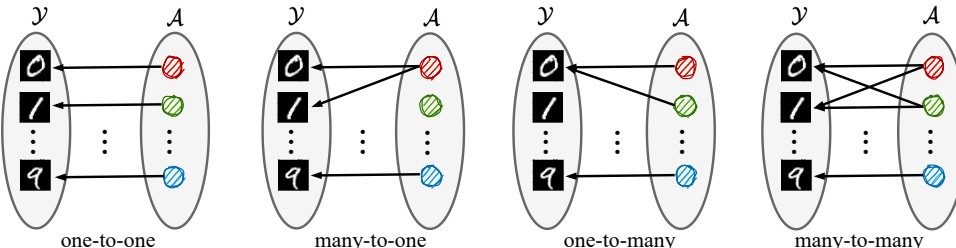

Figure 2: Example of different situations for spurious correlation that can be considered for the colored MNIST dataset. Spurious correlations existed in the majority groups: In the many-to-one situation, multiple digits share the same color; for the many-to-one situation, multiple digits are colored by the same color; Conversely, in the one-to-many situation, a single digit can be colored by several different colors. The many-to-many situation encompasses all of these possibilities, allowing for spurious correlations to occur in multiple directions. Note that the figure only illustrates how labels are spuriously correlated with the attributes.

We show that when $P(a|\mathbf{x})$ follows a one-hot categorical distribution, Eq. 7 is Fisher consistent with maximizing the GBA (Eq. 2). Intuitively, the more likely the combination of $(y, a_\mathbf{x})$ appears in the training dataset, the less we subtract from the original logits. Meanwhile, the less likely the combination of $(y, a_\mathbf{x})$ appears in the training dataset, the more we subtract from the original logits, resulting in less confidence in the prediction. It leads to larger gradients for samples in the minority group, making the network learns more from the minority group. To this end, the logit corrected loss helps reduce the statistical skew. Moreover, Eq. 7 can be further rewritten as

$$\mathcal{L}_{LC}(y, f(\mathbf{x})) = \log\left(1 + \sum_{y' \neq y} e^{f_{y'}(\mathbf{x}) - f_y(\mathbf{x}) + \ln\left(\hat{P}_{y',a_\mathbf{x}}/\hat{P}_{y,a_\mathbf{x}}\right)}\right). \tag{8}$$

It's a pairwise margin loss (Menon et al., 2013), which introduces a desired per example margin $\ln\left(\hat{P}_{y',a_\mathbf{x}}/\hat{P}_{y,a_\mathbf{x}}\right)$ into the softmax cross-entropy. A minority group example demands a larger margin since the margin is large when $\hat{P}_{y,a_\mathbf{x}}$ is small. To this end, LC loss is able to mitigate the geometric skew resulting from maximizing margins. We also empirically compare the training margins of minority groups and majority groups in Figure 4 for different methods. In the next section, we will introduce, given a sample $\mathbf{x}$ in the training dataset, how to estimate $a_\mathbf{x}$ and the group prior.

## 4.2 ESTIMATING THE GROUP PRIOR

In this section, we analyze different spurious correlation relations. The spurious correlation between the target label and the spurious attribute can be categorized into 4 different types (Figure 2): 1) **one-to-one**, where each label only correlates with one attribute value and vice versa; 2) **many-to-one**, where each label correlates with one attribute value, but multiple labels can correlate with the same attribute value; 3) **one-to-many**, where one label correlates with multiple attribute values; and 4) **many-to-many**, where multiple labels and multiple attribute values can correlate with each other. In the next section, we will first discuss the most common one-to-one situation and then extend it to other situations.

### 4.2.1 ONE-TO-ONE

One-to-one is the most common situation studied in the previous works, *e.g.*, the original Colored MNIST dataset. To estimate the group prior probability $P(y, a)$, we consider

$$P(y, a) = \int_\mathbf{x} P(y, a, \mathbf{x})d\mathbf{x} = \int_\mathbf{x} P(y, a|\mathbf{x})P(\mathbf{x})d\mathbf{x} \approx \frac{1}{N}\sum_\mathbf{x} P(y, a|\mathbf{x}). \tag{9}$$

The last approximation is to use the empirical probability estimation to estimate the prior, where $N$ is the number of total samples. It's impractical to use the whole dataset to estimate the group prior after each training iteration is finished. In practice, we try different estimation strategies and find out that

using a moving average of the group prior estimated within each training batch seems to produce reasonable performance (see Figure 3). We apply this method to all the experiments in the paper.

For a training sample of $(\mathbf{x}_i, y_i, a_i)$, since the label $y_i$ is known, we have

$$P(y, a|\mathbf{x}_i) = \begin{cases} P(a|\mathbf{x}_i), & \text{if } y = y_i, \\ 0 & \text{otherwise.} \end{cases} \tag{10}$$

Since the spurious correlation is one-on-one, the number of categories and the number of different attribute values are the same. Without loss of generality, we assume that the $j$-th category ($y^{(j)}$) is correlated with the $j$-th attribute value ($a^{(j)}$), where $j = [1, \ldots, L]$ and $L = K$. $L$ and $K$ are the number of categories and the number of different attribute values respectively.

Since the ERM network would be biased to the spurious attribute instead of the target label, the prediction of the ERM network can be viewed as an estimation of $P(a|\mathbf{x}_i)$. Formally, denote the output logits of the ERM network as $\hat{f}(\mathbf{x}_i)$, and the $j$-th element of $\hat{f}(\mathbf{x})$ is denoted as $\hat{f}_j(\mathbf{x}_i)$, we have

$$P(a = a^{(j)}|\mathbf{x}_i) = \frac{\exp(\hat{f}_j(\mathbf{x}_i))}{\sum_{k=1}^{K} \exp(\hat{f}_k(\mathbf{x}_i))}. \tag{11}$$

Given Eq. 9 to Eq. 11, we can estimate the group prior $P(y, a)$. The associated attribute value in Eq. 5 can be estimated as $a_{\mathbf{x}} = \arg\max_a P(a|\mathbf{x})$.

### 4.2.2  MANY-TO-ONE

Under this scenario, multiple labels can be correlated with the same attribute value. Without loss of generality, we assume that $y^{(1)}$ and $y^{(2)}$ are correlated with $a^{(1)}$, and $y^{(j)}, j > 2$ is correlated with $a^{(j-1)}$. We consider that the first 2 label predictions are spuriously correlated with attribute value $a^{(1)}$, i.e., $\hat{f}_j(\mathbf{x}) \propto P(y^{(j)}, a^{(1)}|\mathbf{x}), j = 1, 2$ and other predictions are similar as the one-to-one situation, where $\hat{f}_j(\mathbf{x}) \propto P(a^{(j-1)}|\mathbf{x}), j > 2$. Compared to the one-on-one mapping, the only difference is the calculation of $P(a = a^{(1)}|\mathbf{x}_i)$ in Eq. 11. Considering both $y^{(1)}$ and $y^{(2)}$ contribute to $a^{(1)}$, we have,

$$P(a = a^{(1)}|\mathbf{x}_i) = P(y^{(1)}, a = a^{(1)}|\mathbf{x}_i) + P(y^{(2)}, a = a^{(1)}|\mathbf{x}_i)$$

$$= [\exp(\hat{f}_1(\mathbf{x}_i)) + \exp(\hat{f}_2(\mathbf{x}_i))] / \sum_{k=1}^{K} \exp(\hat{f}_k(\mathbf{x}_i)). \tag{12}$$

The associated attribute value can then be estimated as well, i.e. $a_{\mathbf{x}} = \arg\max_a P(a|\mathbf{x})$.

### 4.2.3  ONE-TO-MANY

In this scenario, one label can be correlated with multiple attribute values. Since we don't have the attribute label, to distinguish different attributes correlated with the same label, we follow Seo et al. (2022) to create pseudo labels for multiple attribute values. Without loss of generality, we assume $y^{(1)}$ is correlated with $a^{(1)}$ and $a^{(2)}$. Therefore, $P(a = a^{(j)}|\mathbf{x}_i) = \frac{w_j}{w_1 + w_2} \exp(\hat{f}_1(\mathbf{x}_i)) / \sum_{k=1}^{K} \exp(\hat{f}_k(\mathbf{x}_i)), j = 1, 2$, where $w$ is the weight defined in Seo et al. (2022), Eq. (6). The associated attribute value can also be estimated with the estimated posterior.

### 4.2.4  MANY-TO-MANY

Since this is a combination of the previous cases, we can apply the solutions mentioned above together. In order to accurately calculate the prior probability $\mathbf{P}(y, a)$, we need to at least know how the label set $\mathcal{Y}$ and the attribute set $\mathcal{A}$ are correlated. Annotating the category-level relation is much easier than annotating the sample-level attribute. In the case where even the category-level relation is not available, we show in Sec. E that directly applying the one-to-one assumption in other situations (one-to-many and many-to-one) still shows reasonable performance.

### 4.3  GROUP MIXUP

To further mitigate the geometric skew and increase the diversity of the samples in the minority group, we proposed a simple group MixUp method. We also start with the one-to-one situation and other

situations can be derived similarly. Same as Sec. 4.2.1, without loss of generality, we assume the $j$-th category $(y^{(j)})$ is correlated with the $j$-th attribute value $(a^{(j)})$. An training sample $(\mathbf{x}_i, y_i)$ is in the minority group when $\arg\max_{y'} \hat{f}'_y(\mathbf{x}_i) \neq y_i$, else it is in the majority group. Given one sample $\mathbf{x}_i$ in the minority group, we randomly select one sample $\mathbf{x}_j$ in the majority group with the same label $(y_i = y_j)$, instead of using the original $\mathbf{x}_i$ in training, following the idea of MixUp (Zhang et al., 2018), we propose to generate a new training example $(\mathbf{x}'_i, y_i)$ as well as its correction term via the linear combination of the two examples,

$$\mathbf{x}'_i = \lambda \mathbf{x}_i + (1-\lambda)\mathbf{x}_j, \quad \hat{P}'_{y_i,.} = \lambda \hat{P}_{y_i,a^{(i)}} + (1-\lambda)\hat{P}_{y_j,a^{(j)}}, \tag{13}$$

where $\lambda \sim U(0.5, 1)$ to assure that the mixed example is closer to the minority group example. Since both samples are with the same label, we expect their convex combination shares the same label. Using such a convex combination technique increases the diversity of minority groups. The pseudo code of the implementation can be found in Appendix B.

## 5 EXPERIMENTS

In this section, we evaluate the effectiveness of the proposed logit correction (LC) method on five computer vision benchmarks presenting spurious correlations: Colored MNIST (C-MNIST) (Arjovsky et al., 2020), Corrupted CIFAR-10 (C-CIFAR10) (Hendrycks & Dietterich, 2019; Nam et al., 2020), Biased FFHQ (bFFHQ) (Karras et al., 2019; Lee et al., 2021), Waterbird (Wah et al., 2011), and CelebA (Liu et al., 2015). Sample images in the datasets can be found in Figure 5 of Appendix.

### 5.1 EXPERIMENTAL SETUP

**Datasets.** C-MNIST, C-CIFAR-10 and Waterbird are synthetic datasets, while CelebA and bFFHQ are real-world datasets. The above datasets are utilized to evaluate the generalization of baselines over various domains. The C-MNIST dataset is an extension of MNIST with colored digits, where each digit is highly correlated to a certain color which constitutes its majority groups. In C-CIFAR-10, each category of images is corrupted with a certain texture noise, as proposed in Hendrycks & Dietterich (2019). Waterbird contains images of birds as "waterbird" or "landbird", and the label is spuriously correlated with the image background, which is either "land" or "water". CelebA and bFFHQ are both human face images datasets. On CelebA, the label is blond hair or not and the gender is the spurious attribute. The group containing samples of male with blond hair is the minority group. bFFHQ uses age and gender as the label and spurious attribute, respectively. Most of the females are "young" and males are "old".

**Evaluation.** Following Nam et al. (2020), for C-MNIST and C-CIFAR-10 datasets, we train the model with different ratios of the number of minority examples to the number of majority examples and test the accuracy on a group-balanced test set (which is equivalent to GBA). The ratios are set to 0.5%, 1%, 2%, and 5% for both C-MNIST and C-CIFAR-10. For bFFHQ dataset, the model is trained with 0.5% minority ratio and the accuracy is evaluated on the minority group Lee et al. (2021). For Waterbird and CelebA datasets, we measure the worst group accuracy (Sohoni et al., 2020).

**Baselines.** We consider six baselines methods:
- *Empirical Risk Minimization (ERM)*: training with standard softmax loss on the original dataset.
- *Group-DRO* (Sagawa et al., 2019): Using the ground truth group label to directly maximize the worst-group accuracy.
- *Learn from failure (LfF)* (Nam et al., 2020): Using ERM to detect minority samples and estimate a weight to reweight minority samples.
- *Just train twice (JTT)* (Liu et al., 2021b): Similar to LfF but weight the minority samples by a hyperparameter.
- *Disentangled feature augmentation (DFA)* (Lee et al., 2021): Using the generalized cross entropy loss (Zhang & Sabuncu, 2018) to detect minority samples and reweight the minority. Using feature swapping to augment the minority group.

**Implementation details.** We deploy a multi-layer perception (MLP) with three hidden layers as the backbone for C-MNIST, and ResNet-18 for the remaining datasets except ResNet-50 for Waterbirds and CelebA. The optimizer is Adam with $\beta = (0.9, 0.999)$. The batch size is set to 256. The learning rate is set to $1 \times 10^{-2}$ for C-MNIST, $1 \times 10^{-3}$ for Waterbird and C-CIFAR-10, and $1 \times 10^{-4}$ for CelebA and bFFHQ. For $q$ in Eq. 3, it's set to 0.7 for all the datasets except for Waterbird which is set to 0.8. More details are described in Appendix. D.

Table 1: Classification accuracy (%) evaluated on group balanced test sets of C-MNIST and C-CIFAR-10 with varying ratio (%) of minority samples. The baseline method results are taken from Lee et al. (2021) as the same experiment settings are adopted. We denote whether the model requires group or spurious attribute annotations in advance by ✗(i.e., not required), and ✓(i.e., required). Best performing results are marked in **bold**.

| Methods | Group Info | C-MNIST | | | | C-CIFAR-10 | | | |
|---|---|---|---|---|---|---|---|---|---|
| | | 0.5 | 1.0 | 2.0 | 5.0 | 0.5 | 1.0 | 2.0 | 5.0 |
| Group DRO | ✓ | 63.12 | 68.78 | 76.30 | 84.20 | 33.44 | 38.30 | 45.81 | 57.32 |
| ERM | ✗ | 35.19 (3.49) | 52.09 (2.88) | 65.86 (3.59) | 82.17 (0.74) | 23.08 (1.25) | 25.82 (0.33) | 30.06 (0.71) | 39.42 (0.64) |
| JTT | ✗ | 53.03 (3.89) | 62.9 (3.01) | 74.23 (3.21) | 84.03 (1.10) | 24.73(0.60) | 26.90(0.31) | 33.40(1.06) | 42.20(0.31) |
| LfF | ✗ | 52.50 (2.43) | 61.89 (4.97) | 71.03 (2.44) | 84.79 (1.09) | 28.57 (1.30) | 33.07 (0.77) | 39.91 (0.30) | 50.27 (1.56) |
| DFA | ✗ | 65.22 (4.41) | 81.73 (2.34) | 84.79 (0.95) | 89.66 (1.09) | 29.95 (0.71) | 36.49 (1.79) | 41.78 (2.29) | 51.13 (1.28) |
| LC(ours) | ✗ | **71.25 (3.17)** | **82.25 (2.11)** | **86.21 (1.02)** | **91.16 (0.97)** | **34.56 (0.69)** | **37.34 (1.26)** | **47.81 (2.00)** | **54.55 (1.26)** |

Table 2: Worst-group accuracies on Waterbirds, CelebA, and minority-group accuracy on bFFHQ. For the ERM, JTT and Group-DRO baselines, we provide the results reported in Liu et al. (2021a), except for bFFHQ, we rerun the baseline methods. The Group Info column shows whether group labels are available during training.

| Method | Group Info | Waterbirds | CelebA | CivilComments | bFFHQ |
|---|---|---|---|---|---|
| | | Worst | Worst | Worst | Minority |
| Group-DRO | ✓ | 91.4 | 88.9 | - | - |
| ERM | ✗ | 62.9 (0.3) | 46.9 (2.2) | 58.6 (1.7) | 56.7 (2.7) |
| JTT | ✗ | 85.8 (1.2) | 81.5 (1.7) | 69.3 (-) | 65.3 (2.5) |
| LfF | ✗ | 78.0 (0.9) | 77.2 (-) | 58.3 (0.5) | 62.2 (1.6) |
| DFA | ✗ | 87.7 (0.2) | 84.1 (1.2) | - | 63.9 (0.3) |
| LC(ours) | ✗ | **90.5 (1.1)** | **88.1 (0.8)** | **70.3 (1.2)** | **70.0 (1.4)** |

# 6 RESULTS

## 6.1 CLASSIFICATION ACCURACY

Table 1 reports the accuracies on group balanced test sets for all baseline approaches and the proposed method when trained with various minority-to-majority ratios. Models trained with ERM commonly show degraded performance and the phenomenon is aggravated by the decrease in the number of examples in the minority groups. Compared to other approaches, LC consistently achieves the highest test accuracy. The performance gain is even more significant when the minority ratio is low. For example, compared to DFA Sohoni et al. (2020), LC improves the accuracy by 6.03%, 4.61% on C-MNIST and C-CIFAR-10 datasets respectively, when the minority ratio is 0.5%. While at 5% minority ratio, the improvements are 1.51% and 3.42%. It shows the superior performance of the proposed method in datasets with strong spurious correlations. Even compared to the approach which requires the ground-truth attribute label during training (Group DRO), LC still achieves competitive or even better performance. Table 2 shows the model performances on Waterbird, CelebA and bFFHQ datasets. LC again achieves the highest worst/minority-group accuracy among all methods without group information. The clear performance gaps again prove the effectiveness of the proposed method.

Note that all results demonstrated in both Table 1 and 2 are on datasets with one-to-one spurious correlation. We also tested the proposed algorithm on datasets with many-to-one and one-to-many correlations as well. The proposed method also outperforms the best baseline methods. Additional experiments can be found in Appendix E.

## 6.2 ABLATION STUDY

**Effectiveness of each module.** Table 3 demonstrates the effectiveness of the LC loss and the Group MixUp in the proposed method. The evaluation is conducted on the bFFHQ dataset. The first row shows the performance of the baseline ERM network. From rows 2-4, each proposed module helps improve the baseline method. Specifically, adding Group MixUp brings 6.35% of the performance boost, and introducing logit correction is able to improve the performance by 9.64%. Combining both of the elements achieves 12.80% accuracy improvement. Compared our method without Group MixUp (third row in Table 3) to LfF in Table 2, both methods use the same pipeline and the only difference is that we apply LC loss while LfF uses reweighting. The experiment result shows that the proposed LC loss clearly outperforms reweighting (62.2% → 66.51%). The reasons may be due to

the proposed LC loss 1) is Fisher consistent with the balanced-group accuracy; and 2) is able to reduce the geometric skew as well as the statistical skew.

Table 3: Ablation studies on 1) Group MixUp, 2) correcting logit on bFFHQ. Each row indicates a different training setting with ✓mark denoting the setting applied. While correcting the logit individually brings a significant performance boost, adding Group MixUp further improves the performance.

| Group MixUp | Logit Correction | Minority Group Accuracy |
|---|---|---|
| ✗ | ✗ | 56.87 |
| ✓ | ✗ | 63.22 |
| ✗ | ✓ | 66.51 |
| ✓ | ✓ | 69.67 |

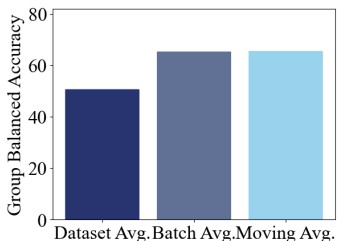

Figure 3: The performance comparison of different strategies for estimating the group prior (on CMNIST with ratio = 0.5%).

**Influence of group prior estimate method.** We test how different group prior estimation strategies in Eq. 9 affect the final performance. We tested 1) updating the prior using all samples in the datasets after finishing each training epoch (Dataset Avg.); 2) updating the prior using all samples in one training mini-batch (Batch Avg.); and 3) keeping a moving average for the batch-level prior (Moving Avg.). The result is shown in Fig. 3. Dataset Avg. performs significantly worse than the other two strategies. This may be because the Dataset Avg. only updates the prior after each epoch. The delay in the prior estimation may mislead the model training, especially in the early training stage when the model prediction can change significantly.

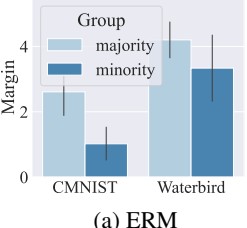
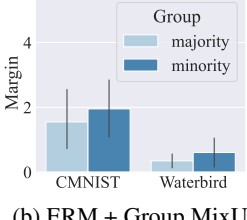
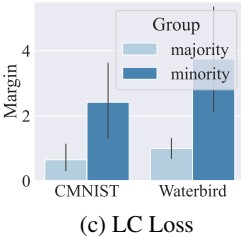

(a) ERM        (b) ERM + Group MixUp        (c) LC Loss

Figure 4: The effect of the proposed logit correction (LC) method on classification margins (defined in Appendix C) on CMNIST and Waterbird datasets. ERM produces a ratio (between the majority group margin and the minority group margin) $> 1$, ERM + Group Mixup has a ratio $< 1$ and the proposed LC loss achieves a ratio $\ll 1$.

**Analysis of training margins.** We show how the proposed LC loss and Group MixUp help reduce the geometric skew. In Sec. 1, we mentioned that a balanced classifier prefers a larger margin on the minority group compared to the margin on the majority group, *i.e.*, the ratio between the majority group margin and the minority group margin should less than 1. In Figure 4, we show the minority group margin and the majority group margin (defined in Appendix C) of the model trained with ERM, LC loss, and ERM + Group MixUp on both C-MNIST and Waterbird datasets respectively. Figure 4 shows that both the proposed LC loss and the Group MixUp can reduce the geometric skew since both of them have a ratio that is less than 1.

## 7 CONCLUSION

In this work, we present a novel method consisting of a logit correction loss with Group MixUp. The proposed method can improve the group balanced accuracy and worst group accuracy in the presence of spurious correlations without requiring expensive group labels during training. LC is statistically motivated and easy-to-use. It improves the group-balanced accuracy by encouraging large margins for minority group and reducing both statistical and geometric skews. Through extensive experiments, The proposed method achieves state-of-the-art group-balanced accuracy and worst-group accuracy across several benchmarks.

ACKNOWLEDGEMENT

SL was partially supported by Alzheimer's Association grant AARG-NTF-21-848627, NSF grant DMS 2009752, and NSF NRT-HDR Award 1922658. CFG acknowledges support from NSF OAC-2103936.

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

APPENDIX

## A  PROOF OF PROPOSITION 1

**Proposition 1.** *Let $P(y, a)$ is the prior on group $(y, a)$, and $P(y, a|\mathbf{x})$ is the true posterior probability of group $(y, a)$ given $\mathbf{x}$, the prediction:*

$$y = \arg\max_{y'} \sum_a \frac{P(y', a|\mathbf{x})}{P(y', a)} = \arg\max_y \sum_a \frac{P(y|a, \mathbf{x})P(a|\mathbf{x})}{P(y, a)}$$

*is the solution to Eq. 2.*

*Proof.* To simplify the notation, we define the output of the classifier as $c = \arg\max_{y' \in \mathcal{Y}} f_{y'}(\mathbf{x})$.

Following Collell et al. (2016), for a group $(y^{(j)}, a^{(k)})$, the accuracy in this group can be written as,

$$Acc(y^{(j)}, a^{(k)}) = \int_{\mathbf{x}} \frac{P(y = y^{(j)}, a = a^{(k)}|\mathbf{x})P(c = y^{(j)}|\mathbf{x})}{P(y = y^{(j)}, a = a^{(k)})} P(\mathbf{x})d\mathbf{x}. \tag{14}$$

GBA in Eq. 2 thus can be rewritten as:

$$GBA = \frac{1}{KL} \int_{\mathbf{x}} \sum_{y^{(j)}} \sum_{a^{(k)}} \frac{P(y = y^{(j)}, a = a^{(k)}|\mathbf{x})P(c = y^{(j)}|\mathbf{x})}{P(y = y^{(j)}, a = a^{(k)})} P(\mathbf{x})d\mathbf{x}. \tag{15}$$

Maximizing Eq. 15 is equivalent to obtain the optimal choice of $P(c = y^{(j)}|\mathbf{x})$ at each $\mathbf{x}$. Since what inside of the integral is

$$\sum_{y^{(j)}} \sum_{a^{(k)}} \frac{P(y = y^{(j)}, a = a^{(k)}|\mathbf{x})P(c = y^{(j)}|\mathbf{x})}{P(y = y^{(j)}, a = a^{(k)})}$$

$$= \sum_{y^{(j)}} \left( \sum_{a^{(k)}} \frac{P(y = y^{(j)}, a = a^{(k)}|\mathbf{x})}{P(y = y^{(j)}, a = a^{(k)})} \right) P(c = y^{(j)}|\mathbf{x}), \tag{16}$$

which is a convex combination, and is maximized at each $\mathbf{x}$ if and only if we place probability 1 to the largest term. That is to say, at each $\mathbf{x}$, we assign 1 to $P(c = y^{(j)}|\mathbf{x})$ where $\sum_{a^{(k)}} \frac{P(y=y^{(j)}, a=a^{(k)}|\mathbf{x})}{P(y=y^{(j)}, a=a^{(k)})}$ is the largest term among all possible $y$ values in $\mathcal{Y}$ and assigning 0 to other terms. Formally,

$$P(c = y^{(j)}|\mathbf{x}) = \begin{cases} 1, & \text{if } y^{(j)} = \arg\max_{y'} \sum_{a^{(k)}} \frac{P(y=y', a=a^{(k)}|\mathbf{x})}{P(y=y', a=a^{(k)})} \\ 0, & \text{Otherwise.} \end{cases} \tag{17}$$

The second equation can be derived by Bayes' theorem. $\qquad \square$

## B  PSEUDO CODE OF THE PROPOSED ALGORITHM

We provide the pseudo-code of the proposed logit correction and Group MixUp in Algorithm 1.

## C  DEFINITION OF CLASSIFICATION MARGIN

Let $f(\mathbf{x}) : \mathbb{R}^d \to \mathbb{R}^k$ be a model that outputs $k$ logits, following previous works (Koltchinskii & Panchenko, 2002; Cao et al., 2019), we define the margin of an example $(x, y)$ as

$$m(x, y) = f(\mathbf{x})_y - \max_{j \neq y} f(\mathbf{x})_j. \tag{18}$$

We can then define the training margin for a group $g = (a, y)$ as the minimum margin of all classes

$$m_g = \min_{i \in g} m(\mathbf{x}_i, y_i). \tag{19}$$

The margins for minority groups and majority groups are defined as the average margin of minority/majority groups.

---

**Algorithm 1:** LC for one-to-one mapping

---

**Input** : Training set $(X, Y)$, Initialize the ERM model $\hat{f}_\theta$ and the robust model $f_\theta$, # epochs $K$, # rampup epoch $T$, moving average momentum $\alpha$.

1 **for** *epoch* = 1 **to** $K$ **do**
2     Sample a mini-batch $\{(\mathbf{x}, y)\}$;
3     Update ERM network $\hat{f}(\theta)$ parameters by training on $\{(\mathbf{x}, y)\}$ with Equation 3;
4     **for** $(\mathbf{x}, y) \in \{(\mathbf{x}, y)\}$ **do**
5        Let $p^{(\mathbf{x}, y)}$ be the ERM model's probability outputs on sample $(\mathbf{x}, y)$.
6        Let $a_{\mathbf{x}} := \arg\max p^{(\mathbf{x}, y)}$ be the estimated value of the spurious attribute.
7        Update the group priors by $\hat{P}_{y,a_{\mathbf{x}}} := \alpha \hat{P}_{y,a_{\mathbf{x}}} + (1 - \alpha) p_{a_{\mathbf{x}}}^{(\mathbf{x}, y)}$
8     **end**
9     (Optional) Perform Group MixUp to obtain the synthesized batch:
10     $\tau := 0.5 \cdot \exp(-5(1 - epoch)/T)^2$ sigmoid ramp up function
11     $\{\mathbf{x}, y, \hat{P}^{(\mathbf{x})}\} = \text{GroupMixup}(\{\mathbf{x}, y, a_{\mathbf{x}}\}, \hat{P}, \tau)$
12     **for** $(\mathbf{x}, y, \hat{P}^{(\mathbf{x})}) \in \{\mathbf{x}, y, \hat{P}^{(\mathbf{x})}\}$ **do**
13        **for** $c$ = 1 **to** *# labeled classes* **do**
14           Correct the $c$th logit of the robust mode by $f(\mathbf{x})_c := f(\mathbf{x})_c + \log \hat{P}_{c,a_{\mathbf{x}}}^{(\mathbf{x})}$
15        **end**
16     **end**
17     Update robust model's parameters $\theta$ with softmax cross entropy loss on $f_\theta(\mathbf{x})$.
18 **end**
19 **Function** `GroupMixup`$(\{\mathbf{x}, y, a_{\mathbf{x}}\}, \hat{P}, \tau)$**:**
20     Obtain a set of samples $\{(\bar{x}, \bar{y}, a_{\bar{x}})\}$ that are estimated to be from minority groups i.e. $y \neq a_{\mathbf{x}}$
21     $\{(\bar{x}, \bar{y}, a_{\bar{x}})\} := \text{shuffle}(\{(\bar{x}, \bar{y}, a_{\bar{x}})\})$;
22     Sample $\lambda \sim \text{Uniform}(1 - 2\tau, 1 - \tau)$;
23     **for** $i, (\mathbf{x}, y, a_{\mathbf{x}}) \in$ *enumerate*$(\{(\mathbf{x}, y, a_{\mathbf{x}})\})$ **do**
24        Sample $(\bar{x}, \bar{y}, a_{\bar{x}})$ from $\{(\bar{x}, \bar{y}, a_{\bar{x}})\}$ such that $y = \bar{y}$
25        $\mathbf{x} := \lambda \mathbf{x} + (1 - \lambda)\bar{x}$
26        $y := y$
27        $\hat{P}^{(\mathbf{x})} := \lambda \hat{P}_{y,\mathbf{x}} + (1 - \lambda)\hat{P}_{y,\bar{x}}$, the correction term for MixUp sample $\mathbf{x}$.
28     **end**
29 **End Function**

---

## D EXPERIMENT DETAILS

We utilize Adam optimizer with $\beta = (0.9, 0.999)$ without weight decay except for CelebA, we set weight decay to $1 \times 10^{-4}$, and a batch size of 256. For Waterbird, we use SGD optimizer with weight decay of $1 \times 10^{-4}$. Learning rates of $1 \times 10^{-2}$, $1 \times 10^{-3}$ and $1 \times 10^{-4}$ are used for Colored MNIST, Waterbird, and CelebA, respectively. We use a learning rate of $5 \times 10^{-4}$ for 0.5% ratio of Corrupted CIFAR-10 and $1 \times 10^{-3}$ for the remaining ratios. We decay the learning rate at 10k iteration by 0.5 for both Colored MNIST and Corrupted CIFAR10. For CelebA, we adopt a cosine annealing learning rate schedule. For Waterbird, we set the $q$ in GCE as 0.8 and 0.7 for other datasets. The ramp-up epoch $T$ is set as 50 for waterbird and CelebA and 2 for other datasets, and the moving average momentum $\alpha$ is set to 0.5 for all datasets.

## E RESULTS ON OTHER SPURIOUS CORRELATION

In the previous section, we report the results on one-to-one mapping which is a common scenario considered by previous works. In this section, we further examine the performance of previous approaches as well as LC on other spurious correlation types.

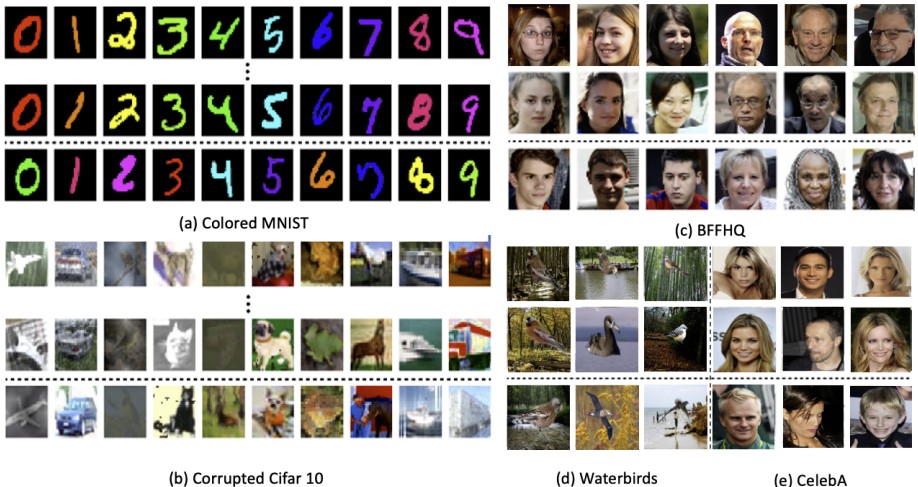

(a) Colored MNIST

(c) BFFHQ

(b) Corrupted Cifar 10

(d) Waterbirds

(e) CelebA

Figure 5: Example images of datasets used in our work. In each dataset, the images above the dotted line demonstrate the majority groups while the ones below the dotted line are minority groups. For Colored MNIST and Corrupted CIFAR-10, each column demonstrates each class.

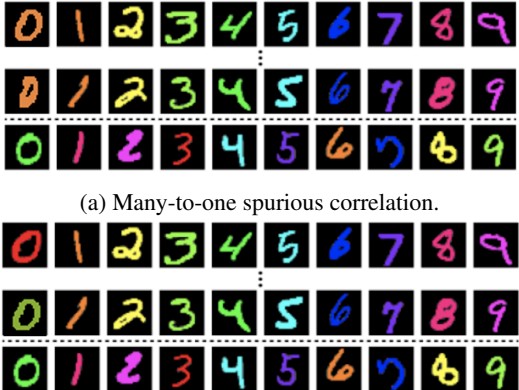

(a) Many-to-one spurious correlation.

(b) One-to-many spurious correlation.

Figure 6: Sample images for datasets containing one-to-many and many-to-one correlations. The first two rows show the training samples from the majority groups and the third row shows the validation set where groups are balanced. For many-to-one, both digits 0's and 1's are colored browns. For one-to-many, digit 0's are colored in different colors (red and dark green).

### E.1 MANY-TO-ONE

The digits in the MNIST training dataset is injected with different colors (similar to the original Colored MNIST). However, we inject the same color into digits 0 and 1 to obtain the many-to-one relationships between the label and the spurious attribute (Figure 6a). We evaluate the accuracy of the proposed logit correction method on the many-to-one setting mentioned in Sec. 4.2.2 (LC+). We also apply LC loss with the one-to-one assumption as LC.

Table 4: Test accuracy on Colored MNIST data with many-to-one correlation.

| Methods | Group Info | | Colored MNIST | | | |
|---------|------------|-----|------|------|------|------|
| | Train | Val | 0.5 | 1.0 | 2.0 | 5.0 |
| ERM | ✗ | ✓ | 31.13 | 50.89 | 57.92 | 82.19 |
| LfF | ✗ | ✓ | 46.22 | 69.32 | 73.33 | 82.57 |
| DFA | ✗ | ✓ | 64.7 | 77.28 | 84.19 | 90.17 |
| LC(ours) | ✗ | ✓ | **65.32** | 78.05 | 84.24 | **90.3** |
| LC+(ours) | ✗ | ✓ | 65.06 | **78.57** | **84.5** | **90.3** |

In Table 4, we report the accuracies on the balanced test set for Colored MNIST. The proposed method (LC and LC+) constantly outperforms all baselines. Although using the exact mapping information shows the best performance (LC+), directly applying the one-to-one assumption shows a very similar performance.

### E.2 ONE-TO-MANY

We augmented the MNIST training dataset with colors (similar to Colored MNIST) except that digit 0 has two colors as its major color attribute, as shown in 2. We evaluate the accuracy of the proposed logit correction method on the one-to-many setting mentioned in Sec.4.2.3 (LC+). We also apply LC loss with the one-to-one assumption as LC.

Table 5: **Benchmark results on the One-to-Many correlation** Test accuracy on Colored MNIST data with one-to-many mapping.

| Methods | Group Info | | Colored MNIST | | | |
|---------|------------|-----|------|------|------|------|
| | Train | Val | 0.5 | 1.0 | 2.0 | 5.0 |
| ERM | ✗ | ✓ | 38.47 | 48.41 | 67.41 | 80.61 |
| LfF | ✗ | ✓ | 52.79 | 66.07 | 75.09 | 83.5 |
| DFA | ✗ | ✓ | 70.11 | 78.75 | 83.06 | 90.41 |
| LC(ours) | ✗ | ✓ | 72.02 | 79.5 | 83.24 | 90.83 |
| LC+(ours) | ✗ | ✓ | **72.26** | **80.1** | **84.1** | **91.25** |

In Table 5, We report the accuracies on the balanced test set for Colored MNIST. The proposed method (LC and LC+) constantly outperforms all baselines. Although using the exact mapping information shows the best performance (LC+), directly applying the one-to-one assumption shows a very similar performance.

## F MORE ABLATION STUDY

### F.1 ABLATION ON $q$

We conduct an ablation study to better understand $q$ in the GCE loss. Intuitively, when $q$ is closer to 0, it results in a loss function with a gradient that emphasizes the hard example (e.g. cross-entropy), and when $q$ is closer to 1, it results in a loss closer to mean absolute error which produces a gradient that emphasizes less on hard examples. The choice of $q$ depends on different datasets of how fast the easy examples (i.e. the spurious correlated example) can be learned. We conducted an ablation study on the waterbird dataset, it turns out the proposed method is quite robust to different values of $q$, as illustrated in Table 6.

Table 6: Ablation study on $q$ of GCE.

| $q$ | 0.1 | 0.3 | 0.5 | 0.7 | 0.8 | 0.9 |
|-----|-----|-----|-----|-----|-----|-----|
| LC | 84.2 | 88.6 | 88.9 | 89.9 | 90.7 | 89.4 |

### F.2 Logit correction v.s. reweighting

We also conduct experiments to compare logit correction with re-weighting/re-sampling. There is both theoretical and empirical evidence showing that LC is more effective than reweighting for training a linear classifier since it is not only Fisher consistent, but also able to address the geometric skew. Theoretically, recent papers Xu et al. (2020); Sagawa et al. (2020) proved that overparametrized linear model, regardless of trained with reweighted cross entropy and original cross-entropy, would eventually result in the max-margin classifier with enough training. This is also validated by our experiments in which we adopt an ImageNet pre-trained ResNet-18, and only train the last linear classifier on Corrupted CIFAR10 (5%), we obtain that reweighting with the group prior (GBA: 23.75%) results in slightly better results than ERM (GBA: 20.77%). Both are worse than logit correction (GBA: 30.20%).

We further compare reweighting with logit correction on the fully trained model when ground truth group information are available. LC also outperforms the Fisher-Consistent reweighting (Table 7).

Table 7: Comparison of reweighting and logit correction with ground truth group information.

| Method | ERM | Fisher-Consistent Reweighting | logit correction |
|--------|-----|-------------------------------|------------------|
| Test acc. | 39.51 | 42.13 | 54.31 |

