# OpenReview forum: "Avoiding spurious correlations via logit correction"
_ICLR.cc/2023/Conference — ICLR 2023 poster_

### Official Review · Reviewer_tcpr · 2022-10-21

**Confidence:** 3
**Correctness:** 4
**Technical Novelty And Significance:** 3
**Empirical Novelty And Significance:** 3
**Recommendation:** 6

**Clarity, Quality, Novelty And Reproducibility:**

I found the paper to provide an interesting solution to an important problem in real world ML applications. To the best of my knowledge the presented work is novel.

As discussed in the previous section, the clarity needs to be improved.


**Strength And Weaknesses:**

**Strengths**
1. Spurious correlations are very common in real data. This paper proposes an interesting solution to learning classifiers that can avoid them, importantly without assuming that the spurious attribute is known during training

2. The LC loss is theoretically grounded, since as the authors show minimizing the logit-correction loss is equivalent to maximizing the group-balanced accuracy

3. The proposed method outperforms competing ones in popular benchmarks for the task. Ablation studies show the importance of all the novel components introduced in the paper.


**Weaknesses**

_Clarity_

The logic correction term is the key ingredient of the proposed method. As such, from section 4.1 it should be obvious and very clear to the reader how the correction term is computed and why it works, which I believe it is not the case.

1. While Equations (4) -> (7) provide all the mathematical steps in detail, the authors should provide the reader intuition on what each of those equations represent. I really liked the simple waterbird example from Figure 1, and I believe it could be used as a running example throughout section 4.1 to give a more concrete intuition of the formula and assumptions presented in the section.

2.  $\Delta_{y, a_x}$ is never even defined explicitly with a formula.

3. Section 4.2 would also greatly benefit from providing some intuition to the reader


_Applicability to more realistic datasets_

1. While the authors perform experiments on standard benchmarks, I would have liked more real-world experiment, to make sure that the proposed method generalizes to more realistic tasks where the spurious attributes might be very subtle and hard to infer (as is the case for example in medical imaging datasets).

2. I have some doubts that the linear combination assumption of the group mixup will work in a real dataset with higher dimensional images and objects that might have different scale/positions. Would you expect it to generalize well to more realistic tasks?





**Summary Of The Paper:**

This paper presents a loss function for classification that allows to mitigate the issue of spurious correlations in data where there is a majority group in which correlations are present, and a minority group without spurious correlations.

The proposed method consists of training two networks: 1) a ERM network that uses the generalized cross entropy loss, and that is therefore biased to be better at predicting elements containing spurious correlations (in the majority group), 2) a robust network trained with the standard cross entropy loss, but after having scaled the logits by a correction term produced from network 1.

Experiments show that the method is able to reduce geometry and statistical skews, and outperforms competing method in a number of standard benchmarks.



**Summary Of The Review:**

The presented method is interesting and has potential, but in its current state the impact of this paper is limited by the lack of clarity in the theoretical section.

---

> ### Author Response · Authors · 2022-11-19
> **Author Response**
>
> We thank the reviewer for providing detailed paper writing suggestions. We will revise our paper accordingly in the final version.
>
> >**Q1**: Provide more real-world experiments. Does the algorithm work with very subtle spurious attributes?
>
> **A1**: As also suggested by other reviewers, we provided additional results on the CivilComments, a real-world NLP dataset. The proposed algorithm also achieves state-of-the-art performance (please refer to Q1 in replies to common questions). For subtle attributes, most of the unsupervised spurious correlation mitigation methods require that the spurious attribute needs to be strong enough to be "picked up" by the ERM classifier. Our algorithm follows the same assumption.
>
> >**Q2**: Would you expect group MixUp to generalize well to more realistic tasks?
>
> **A2**: Same as the original MixUp idea, the goal of group MixUp is not to create semantically meaningful samples. Instead, it acts as a regularizer to create more margin for the minority group. Since the original MixUp paper works on realistic datasets where the objects have different scales and positions (such as ImageNet), we expect the proposed group MixUp will also generalize to similar datasets.

---

> > ### Comment · Reviewer_tcpr · 2022-11-21
> > **Leaning towards acceptance**
> >
> > Thanks for the reply and the additional experiment. Having also read the other reviews and rebuttals, I am still leaning towards acceptance but will not raise the score.
> >
> > As also noted by other reviewers, I strongly believe clarity still needs to be improved, but I don't see any change to the paper.

---

> > > ### Author Response · Authors · 2022-12-01
> > > **Thank you for leaning acceptance**
> > >
> > > We edited the paper and will make sure the improvements in clarity, as you have suggested, are reflected in the revised version of the paper. Thank you!

---

### Official Review · Reviewer_aF7f · 2022-10-25

**Confidence:** 3
**Correctness:** 3
**Technical Novelty And Significance:** 4
**Empirical Novelty And Significance:** 3
**Recommendation:** 6

**Clarity, Quality, Novelty And Reproducibility:**

In general, the paper and appendix are both quite clear.
I believe that the technique is sufficiently novel and that necessary details have been provided to ensure reproducibility.

Will the code be made available after acceptance?

**Nitpicks**

* I found the short-hand in eq. 2, $\mathbf{P}_{\mathbf{x}|(y, a)}(\cdot)$, much less clear than the full expression in the appendix.
* Unclear without Fig. 1 whether GCE loss is applied after softmax.
* It seems like $f^*$ is introduced and defined in equation 4, however this is not clear.
* The set notation in Algorithm 1 seems incorrect (e.g. for-loops, shuffle).
* LC was described as Logit Correlation in one place (not Correction)
* Some grammar issues (associates, assumes, exist)
* Some verbs mis-used (cooperating, explored)

**Strength And Weaknesses:**

**Strengths**

1. Based on the work of Menon et al., logit correction seems like a highly effective tool for enforcing different margins from different classes in deep learning. Using this to address spurious correlations makes sense.

1. Both Logit Correction and Group Mixup made a significant contribution to the improvement in results.

1. The inference rule in Proposition 1 was unfamiliar to me and the derivation seems good.

1. Empirical evaluations show the method to be highly effective across all datasets.

1. Algorithm 1 helps to understand the method.

**Weaknesses: Experimental**

1. The ablative study was only conducted for one dataset. It would be better if these 3 ablations were added as rows in Tables 1 and 2 for all datasets.

1. DFA missing from Table 2, JTT missing from Table 1.

1. No confidence intervals (standard deviations) in Tables 1 and 2, even for experiments on relatively small datasets (MNIST, CIFAR). This could be especially important for the "Worst Group" metrics, which would probably higher variance than the mean-based metrics.

1. It is stated that the ERM results in Tables 1 and 2 were taken from past papers. To ensure an apples-to-apples comparison, it would be better to run at least the ERM baseline in the exact same pipeline, only disabling your features. (Like the first row of Table 3, but for all the datasets in Tables 1 and 2? Note that Table 3 differs from Table 2, i.e. 56.87 vs 55.47 for MGA with bFFHQ.)

**Weaknesses: Method and text**

1. Was it not necessary to stop the gradients from the ERM branch to the Robust branch?

1. For continuous attributes, it seems like the LC loss could still be used, but Group Mixup would not be directly applicable.

1. The use of the predicted label as the correlated attribute is quite a specific choice (line 6 of Algorithm 1 in appendix). This attribute will always be one-to-one, with ground-truth and predicted labels coinciding in the majority groups. I feel like this particular choice should be given more attention in the formulation. Otherwise, the method seems much more general than the empirical evaluation, since IIUC all experiments in the main paper use this specific configuration. (Only Appendix E contains results that do not.)

1. When the attribute is not the predicted label, it seems necessary to specify which attribute values are correlated with which class labels (Sections 4.2.2, 4.2.3). This is not necessarily a weakness, as sometimes it may be useful to specify this manually. However, it seems that it might also be automatically determined from a class-attribute confusion matrix? How would the accuracy be affected if it was specified incorrectly?

1. It's not clear whether logit correction being more effective than re-weighting/re-sampling is specific to deep learning. For example, Kang et al. (ICLR 2019; "Decoupling Representation and Classifier for Long-Tailed Recognition") found that it was best to learn features using instance-balanced sampling and then learn a linear classifier using class-balanced sampling. Does the LC loss still improve the results when training a linear classifier with frozen features?

1. It is hypothesized that the effectiveness of the LC loss is due to being Fisher consistent. However, it seems that re-weighting samples rather than adjusting the margin might also be Fisher consistent (w.r.t. the group-balanced distribution)? If this is not the case, can you state this in the paper and provide a reference?

**Uncertain**

I'm not an expert on spurious correlations, so I'm not sure whether the choice of datasets is appropriate, in particular the use of datasets with only two classes. I'll defer to other reviewers here.

**Summary Of The Paper:**

This paper considers the problem of spurious correlations in deep learning, using image classification as a case study.
This problem arises when some attribute can be observed in the training set that is correlated with the class label, however the correlation does not hold in general.
The formulation assumes that there is one discrete spurious "attribute" variable, for which we know (or can determine) the values that frequently co-occur with each class.
In practice, this is generally the possibly-incorrect class predicted by a model trained with an ERM loss.
The proposed method comprises two components: Logit Correction loss and Group Mixup.
Similar to logit correction for class-imbalanced learning, the LC loss seeks a greater margin for examples in the less frequent groups, where groups are defined as a (class, attribute) pair.
Group Mixup replaces each minority example with a convex combination of itself and a majority example with the same label, mixing both the inputs and the logit corrections.
The method is compared to several baselines on both synthetic and realistic datasets.

**Summary Of The Review:**

Overall, the approach is well-motivated and the results seem strong. However, I have some concerns about the justifications and the evaluation. Also, it seems that all experiments in the main paper use a particular choice of attribute, and are thus less general than the formulation. I'm leaning towards accept, but I may downgrade my rating if my concerns are not addressed.

---

> ### Author Response · Authors · 2022-11-19
> **Author Response (1/2)**
>
> >**Q1**. The ablative study was only conducted for one dataset. It would be better if these 3 ablations were added as rows in Tables 1 and 2 for all datasets.
>
> **A1**: Due to time constraint, we add an ablation study on CivilComments. We will add more in the appendix of the final paper.
> | Group Mixup   |  LC   | Worst Acc. |
> | --            | ----  |  ----      |
> | No            |   No  |    58.2    |
> | Yes           |   No  |    62.2    |
> | No            | Yes   |    67.9    |
> | Yes           | Yes   |    70.3    |
>
> >**Q2**: Add DFA to Table2 and JTT to Table 1
>
> **A2**:
> Please refer to Q2 in replies to common questions.
>
> >**Q3**: Add confidence intervals to Table 1 and 2.
>
> **A3**: Please refer to Q2 in replies to common questions.
>
>
> >**Q4**: Rerun ERM baseline using the same pipeline as the proposed method in Table 1 and Table 2.
>
> **A4**: Since the proposed method adopts the same pipeline of running ERM as DFA, we directly use their number for Table 1. We rerun ERM for Table 2, and we will update the results to Table 2. The performances are similar on CelebA (46.92 vs. 47.2) and bFFHQ (56.67 vs. 55.47). In Waterbirds, the self-run result is worse than the public number (62.85 vs. 72.6).
> | Method     |  Waterbirds   | CelebA |   bFFHQ |
> | --         | ----          |  ----  |  ----   |
> | ERM        |  62.85(0.30)   |  46.92 (2.17) |   56.67 (2.66)  |
>
>
> >**Q5**: Was it not necessary to stop the gradients from the ERM branch to the Robust branch?
>
> **A5**: It's necessary. From the implementation point of view, we stop the gradient since we adopt a moving average to smooth out the estimation across iterations. Stopping the gradient would make backpropagation easier. Intuitively, if we don't stop the gradient, the two networks would be more symmetric (i.e. they both learn from the same loss). It is undesirable since we would like two networks to be divergent so one would capture the bias and another branch can be more robust.
>
> >**Q6** For continuous attributes, it seems like the LC loss could still be used, but Group Mixup would not be directly applicable.
>
> **A6** Unfortunately, the proposed algorithm is not trivial to be extended to continuous attributes. We need to redefine the group prior in that case. We would leave this as our future work.
>
> >**Q7**: Is one-to-one mapping required for the proposed algorithm?
>
> **A7**: It's not required. Although knowing the exact relation between the label and the attribute will give us the best estimation of the prior distribution, as mentioned by the reviewer, we empirically showed in Appendix E that not using the exact mapping (e.g., using one-on-one as the assumption for a many-to-one case) also gives us compatible performance. Experiments on the CivilComments (Q1 in replies to common questions) also show a good performance on one-on-one mapping. We thank the reviewer pointing this out and we also believe the proposed method can be more general. However, we don’t have a concrete conclusion at this point. We will add the discussion in our final version and leave the generalization as our future work.

---

> > ### Author Response · Authors · 2022-11-19
> > **Author Response (2/2)**
> >
> > >**Q8**: It seems necessary to specify which attribute values are correlated with which class labels in Sec 4.2.2 and 4.2.3. Can it automatically be determined by the confusion matrix?
> >
> > **A8**: We believe there are some misunderstandings here. In Sec 4.2.2 and 4.2.3, we don't assume knowing the exact mapping between the target label and the attribute values. Instead, the proposed algorithm only requires knowing the mapping relations. For example, in CMNIST, the proposed algorithm only requires knowing that number “1” and number “2” are correlated to one color and it’s not necessary to know if the color is red or yellow. For the confusion matrix, since the attribute value is unknown during the training process, it may not be possible to calculate the confusion matrix.
> >
> > >**Q9**: It's not clear whether logit correction is more effective than re-weighting/re-sampling in training a linear classifier.
> >
> > **A9**: There are both theoretical and empirical evidences showing that LC is more effective than reweighting for training a linear classfier, since it is not only Fisher consistent, but also able to address the geomeric skew. Theoratically, recent papers [2,3] proved that overparametrized linear model, regardless trained with reweigted cross entropy and original cross entropy, would eventually result in the max-margin classifier with enough training. This is also validated by our experiments in which we adopt a ImageNet pretrained resnet 18, and **only train the last linear classifier** on Corrupted CIFAR10 (5\%), we obtain that **reweighting with group prior （GBA：23.75\%） results in slightly better results than ERM (GBA: 20.77\%). Both are worse than logit correction (GBA: 30.20\%).
> > [2] Byrd et. al. What is the Effect of Importance Weighting in Deep Learning?
> > [3] Sagawa et. al. An Investigation of Why Overparameterization Exacerbates Spurious Correlations.
> >
> >
> >
> > >**Q10**: It is hypothesized that the effectiveness of the LC loss is due to being Fisher consistent. Re-weighting samples rather than adjusting the margin might also be Fisher consistent.
> >
> > **A10**: The effectiveness of the LC loss is due to 2 facts: 1) it’s Fisher consistent; and 2) it helps mitigate both the statistical skew and the geometric skew (Sec. 4.1). It’s true that if we carefully select the weight, reweighting can also be Fisher consistent（e.g. proportional to the number of examples in each group). However, since reweighting only helps mitigate the statistical skew, even if it’s Fisher consistent, its performance would be still worse than the proposed LC loss. We experimentally verify this on Corrupted CIFAR-10 (5\%). LC performs the best:
> > | Method     | ERM   | Fisher-Consistent ReWeighting |   logit correction |
> > | --         | ----          |  ----  |  ----   |
> > |  Test acc.     |  39.51   | 42.13 |   54.31  |
> >
> >
> > We will release the code and follow your suggestions to improve the paper writing in the final version.

---

> > > ### Comment · Reviewer_aF7f · 2022-11-21
> > > **Still leaning accept**
> > >
> > > Thank you, authors, for a thorough rebuttal.
> > >
> > > The ablative experiments with CivilComments are convincing, and I trust that the complete ablative experiments would be included in a final version.
> > >
> > > It's interesting that logit correction is significantly better than simple reweighting when training even a linear classifier (C-CIFAR10-5%). Thanks for running this experiment.
> > >
> > > It seems that the attribute formulation is still somewhat unclear to myself and other reviewers. Perhaps it would help to have clear formulae for $p(y, a | x)$ or $\Delta_{y, a}$ as well as $\hat{a}$ in each of the subsections of 4.2.
> > >
> > > I believe that the method is sufficiently different from LfF, which uses reweighting rather than logit adjustment and does not consider groups explicitly. It would be illuminating to compare with a version of LfF that instead uses logit adjustment.
> > >
> > > I found another relevant reference to include: Kini et al. "Label-Imbalanced and Group-Sensitive Classification under Overparameterization" (NeurIPS 2021). This paper compares Group Logit Adjustment and Group Vector Scaling losses on the "Waterbirds" problem (with ground-truth group annotations, I believe).
> > >
> > > I'm still leaning towards accept. Please address all issues around clarity and missing relevant references (e.g. LISA) in any final version.

---

> > > > ### Author Response · Authors · 2022-12-01
> > > > **Thank you for leaning acceptance**
> > > >
> > > > Thank you for leaning acceptance. Based on your comments on the clarity of sections 4.1 and 4.2. We have updated the original notation $\Delta_{y,a_x}$ to $\hat{P}_{y,a_x}$, which better refers to the estimation of the group prior $P(y,a_x)$, we have also rephrased the flow from Eq 5 - 7, and addressed other clarity issues, which will be reflected in the final version. Thank you for providing another reference as well, we will add it to our paper.

---

### Official Review · Reviewer_H482 · 2022-10-25

**Confidence:** 3
**Correctness:** 3
**Technical Novelty And Significance:** 3
**Empirical Novelty And Significance:** 3
**Recommendation:** 6

**Clarity, Quality, Novelty And Reproducibility:**

The paper is generally clear and well-written. The logit correction loss function is novel in this setting to the best of my knowledge, though the originality of the paper is limited by weakness #1 above. The reproducibility of the paper is unknown as the authors have not included their code in the supplementary, though they do provide implementation details in the paper and Appendix D.

**Strength And Weaknesses:**

Strengths:
- The paper is generally well-written.
- The proposed method outperforms the baselines on standard benchmark datasets.


Weaknesses:
1. The main weakness of the paper is the novelty of the proposed method, which is limited by two factors:
- The proposed group MixUp method is nearly identical to intra-label LISA [1], which the authors do not reference. It is also similar to [2], which I do not expect the authors to have referenced as it is too recent.
- The GCE loss is the same as the loss used in training the biased model in Learning from Failure, which the authors also do not mention.

2. It seems that, for the method to work, the user needs to know in advance which of the four spurious attribute to label mappings is present within the data. In particular, in Sections 4.2.2-4.2.4, it seems that we would need to know the values of the spurious attribute that are correlated with each value of the label. If so, this seems like a major drawback of the method. The authors should discuss this further, and perhaps conduct experiments in the case where the mapping is misspecified (e.g. we assumed it was one-to-one but it was actually many-to-one).

3. The authors should show the performance of the estimation of the group prior over training. Does this converge to the real values?

4. The authors should evaluate their method on the CivilComments dataset, which is standard in the spurious correlation setting. This might be a tricky setting for the method to define the spurious attribute to label mapping.

5. For transparency, the authors should show the performance of more recent methods such as CNC [3]. They should also show the performance of all methods on all datasets (for example, JTT is missing in Table 1, and there are a couple missing from Table 2).

6. The authors should conduct an experiment showing the effect of varying $q$ in the appendix. Why have they been set to 0.7 and 0.8?

[1] https://arxiv.org/abs/2201.00299
[2] https://arxiv.org/abs/2209.08928
[3] https://arxiv.org/abs/2203.01517

**Summary Of The Paper:**

The authors tackle the spurious correlation problem in the setting where spurious attribute values are unknown. They propose a method where a two-branch neural network is trained, one branch with the generalized cross-entropy, and the other with a logit correction loss which depends on estimated class priors computed from the first branch. Unlike prior two-stage methods like JTT, the proposed method only requires training a single network. The evaluate their method on standard spurious correlation datasets, finding that it improves over the baselines.

**Summary Of The Review:**

Due to inherent issues with the method (Weaknesses #1-2) and issues with the empirical evaluation (Weaknesses #3-6), I recommend rejection at this time pending the authors' rebuttal.

Post-rebuttal update: The authors have addressed my major concerns with the empirical evaluations through new experiments. I am now leaning towards accept.

---

> ### Author Response · Authors · 2022-11-19
> **Author Response**
>
> >**Q1**: Novelty is limited from 2 fronts: 1) Group MixUp method is nearly identical to intra-label LISA; 2) The GCE loss is the same as LfF.
>
> **A1**:
> - We thank the reviewer for pointing out the references. We will cite both papers in our final version. Both intra-label LISA and group MixUp are direct extensions of MixUp. The main difference is that intra-label LISA extends MixUp into domain shift problems where domain labels/information are known, while the group MixUp extends MixUp to spurious correlation problems where group information is unknown.
> - As we mentioned in the paper, the GCE loss is first proposed by Zhang & Sabuncu (2018) and we directly apply it in our framework. It is not our claimed contribution.
>
> We want to again highlight our contributions. First, we proposed the logit correction loss which alleviates both statistical and geometric skews and is proved to be Fisher Consistent with GBA. Second, we propose the group MixUp to further mitigate geometric skew. Finally, extensive experiments show the effectiveness of the proposed method.
> >**Q2**: The authors should discuss this further, and perhaps conduct experiments in the case where the mapping is misspecified (e.g. we assumed it was one-to-one but it was actually many-to-one).
>
> **A2**:  The misspecified cases are discussed in Appendix E, where we empirically show that although knowing the exact mapping leads to the best performance, misspecifying the mapping type (assuming one-to-one in many-to-one situation (Appendix E.1) or one-to-many situation (Appendix E.2)) also achieves compatible performance. We also add the table here:
> #### Many-to-One Mapping
> | Methods | Group Info | C-CMNIST 0.5 | C-CMNIST 1.0 | C-CMNIST 2.0 | C-CMNIST 5.0 |
> |---------|------------|:------------:|:------------:|:------------:|:------------:|
> | ERM     |     No     | 31.13        | 50.89        | 57.92        | 82.19        |
> | LC (one-to-one)      |     No     | 65.32        | 78.05        | 84.24        | 90.3         |
> | LC (many-to-one)    |     No     | 65.06        | 78.57        | 84.5         | 90.3         |
>
> #### One-to-Many Mapping
> | Methods | Group Info | C-CMNIST 0.5 | C-CMNIST 1.0 | C-CMNIST 2.0 | C-CMNIST 5.0 |
> |---------|------------|:------------:|:------------:|:------------:|:------------:|
> | ERM     |     No     | 38.47        | 48.41        | 67.41        | 80.61         |
> | LC (one-to-one)      |     No     | 72.02        | 79.5         | 83.24        | 90.83        |
> | LC (one-to-many)    |     No     | 72.26        | 80.1         | 84.1         | 91.25        |
>
> >**Q3**: The authors should show the performance of the estimation of the group prior during training. Does this converge to the real values?
>
> **A3**: The estimation of the group prior converges to a value close to the ground truth value. For example, on the water birds dataset, the estimation converges to
> |    -  | Land  |Water       |
> | -------  | ----------- |   ----------- |
> | Landbird      |  0.93      | 0.163     |
> | Waterbird      |   0.070    | 0.837    |
>
> and the *ground truth* prior calculated using group labels is:
> |    -  | Land  |Water       |
> | -------  | ----------- |   ----------- |
> | Landbird      |  0.984      | 0.148     |
> | Waterbird      |   0.016    | 0.852    |
> >**Q4**: The authors should evaluate their method on the CivilComments dataset. How to define the spurious attribute to label mapping?
>
> **A4**: The proposed LC outperforms JTT and CNC on CivilComments. We *simply define the mapping as one-to-one mapping*, i.e. we assume different attributes in CivilComments form a single spurious attribute and correlated with the label. For the performance table, please refer to Q1 in replies to common questions.
> >**Q5**: Show the performance of all methods on all datasets
>
> **A5**: please refer to Q2 in replies to common questions.
> >**Q6**: Showing the effect of q. Why is q set to 0.7 and 0.8?
>
> **A6**: We conduct a grid search for the best value of $q$ on the validation set, and the performance tables in the paper show the best accuracy. Intuitively, when q is closer to 0, it results in a loss function with a gradient that emphasizes the hard example (e.g. cross-entropy), and when q is closer to 1, it results in a loss closer to mean absolute error which produces a gradient that emphasizes less on hard examples [1]. The choice of q depends on different datasets of how fast the easy examples (i.e. the spurious correlated example) can be learned. **We conducted an ablation study on $q$ on the waterbird dataset, it turns out the proposed method is quite robust to different values of $q$**.
> | q   |  0.1 | 0.3 | 0.5 | 0.7 | 0.8 | 0.9 |
> | --  | ---- | --- | --- | --- | --- | --- |
> | LC worst group acc. |0.842 |0.886|0.889|0.899|0.907|0.894|
>
> [1] Wang et. al., IMAE for Noise-Robust Learning: Mean Absolute Error Does Not Treat Examples Equally and Gradient Magnitude’s Variance Matters
> >**Q7**：Reproducibility
>
> **A7**: We will release our code upon acceptance.

---

> > ### Comment · Reviewer_H482 · 2022-11-21
> > **Thanks for the response!**
> >
> > I thank the authors for their detailed response. The new results on CivilComments and the convergence to the group priors are convincing evidence that the method is working as intended. In addition, the ablation studies (on $q$, and on the mapping misspecification) seem to demonstrate that the method is quite robust to different hyperparameter settings. The authors should be sure to add these results to the final revision of the paper.
> >
> > All of my major concerns have been addressed. I am now leaning towards accept, and have increased my score as a result.

---

> > > ### Author Response · Authors · 2022-12-01
> > > **Thank you for your positive feedback**
> > >
> > > It is encouraging to see that we are able to address your concerns, we appreciate that you increased your score. Thank you!

---

### Official Review · Reviewer_5Aoq · 2022-10-25

**Confidence:** 3
**Correctness:** 2
**Technical Novelty And Significance:** 3
**Empirical Novelty And Significance:** 2
**Recommendation:** 6

**Clarity, Quality, Novelty And Reproducibility:**

Although the paper provides some insights and good empirical evidence, the clarity can be much improved. I shall suggest shifting the algorithmic block to the main text to better motivate the algorithm. I believe the work is somewhat related to LfF. However, the technical contributions are novel to the best of my knowledge.

**Strength And Weaknesses:**

Strengths:
1. The proposed method is simple to implement.
2. The authors provided theoretical justification.
3. The study is supported through detailed empirical analysis and ablations

Weakness
1. I believe the proposed approach shares some similarities with LfF which also uses a two-branch network with generalized cross-entropy loss for mitigating spurious correlations. The authors should provide a detailed discussion of the difference between the two papers.

2. The average accuracy is not reported in Table 1 and 2. Further, the authors also do not report variances over multiple runs, which slightly diminishes the trustworthiness of the results.

3. According to me, writing can be improved. There are some grammatical mistakes that make it hard to understand the concept in some places. (Minor)

Questions:
1. I am confused about how you get the samples from minority groups while doing GroupMixup if the group labels are not assumed. Further, to the best of my knowledge, both JTT and LfF use group-labeled validation sets for hyperparameter tuning. It's not very clear why the authors indicated that they do not require group information.

2. Can this proposed method be also applied for spuriously correlated NLP datasets such as MultiNLI and CivilComments. It would be interesting to see the performance.



**Summary Of The Paper:**

This paper tackles the problem that deep neural nets trained for classification learn spurious correlations that can negatively impact generalization. For this purpose, the authors propose Logit Correction (LC) loss to mitigate the effects of spurious correlations. Further, to synthesize more samples from the minority groups, the authors propose Group MixUp.


**Summary Of The Review:**

Although the contributions are interesting, according to me the paper needs to be refactored to improve the flow and in its current shape, the overall impression is not positive.

---

> ### Author Response · Authors · 2022-11-19
> **Author Response**
>
> > **Q1**: The proposed approach shares some similarities with LfF which also uses a two-branch network with generalized cross-entropy loss for mitigating spurious correlations.
>
> **A1**: As we mentioned in Sec. 2, although LfF also applies ERM and a 2-branch structure to detect samples in the minority groups, it mainly takes the advantage of a heuristic reweighting to address the statistical skew. **The heuristic reweighting strategy proposed in LfF is not guaranteed to be fisher consistent with Group-Balanced Accuracy (GBA) and has little impact on mitigating the geometrical skew. We proposed logit correction extending from logit adjustment, which is not only Fisher consistent with GBA but also has the capability to mitigate the geometrical skew.** In Sec 6.1, we show that the proposed method outperforms LfF by a clear margin for all the benchmarks. We will highlight the difference in the revised version.
>
> >**Q2**: Please report the average accuracy and the variance of the proposed method over multiple runs.
>
> **A2**: Please refer to Q2 in replies to common questions.
>
>
> >**Q3**: How you get the samples from minority groups while doing GroupMixup if the group labels are not assumed.
>
> **A3**: As we defined in Sec. 4.3, a training sample is considered to be a sample in the minority group if its predicted label from the ERM network is different from its target label. The motivation behind is, we assume that the ERM network would be biased to the spurious attribute instead of the target label (Sec. 4.2.1). Therefore, the inconsistency between the target label and the ERM prediction indicates the inconsistency between the target label and the spurious attribute (the definition of samples in the minority group, Sec. 1).
>
> >**Q4** It's not very clear why the authors indicated that JTT and LfF do not require group information.
>
> **A4** Sorry for the confusion, the Group info column in both Tables indicates whether a method uses group info during training or not. JTT and LfF do not use group information for model training but use it during the hyperparameter tuning. We will clarify this in the final version.
>
> >**Q5**: What’s the performance of the proposed model on spuriously correlated NLP datasets such as CivilComments.
>
> **A5**: Please refer to Q1 in reply to common questions.
>
> We will follow the suggestions provided by the reviewer to improve the clarity of the paper.

---

> > ### Author Response · Authors · 2022-12-01
> > **Thank you for increasing the score**
> >
> > Dear Reviewer 5Aoq,
> >
> > We are encouraged that you raised the score, and hopefully, we have addressed your concerns, please let us know if you have other questions. Thank you!

---

### Author Response · Authors · 2022-11-19
**Reply to common questions**

We thank the reviewers for their time and encouraging feedback. We are encouraged that all the reviewers find the proposed method interesting, effective, and technically novel, supported through “detailed empirical analysis and ablations” (Reviewer 5Aoq);  Reviewer tcprv also pointed out that the method is “theoretically grounded”; Reviewer aF7f  believes that the method is “highly effective across all datasets”; We would like to respond the common questions raised by multiple reviewers here and then respond to each of the reviewers in detail.

>**Q1** Can this proposed method be also applied for spuriously correlated NLP datasets e.g., CivilComments?

**A1** We evaluate our model on CivilComments dataset. The proposed LC outperforms JTT and CNC on CivilComments (Table below). Since the mapping is unknown, we simply adopt one-to-one mapping in this experiment.
| Method   |  ERM  | LfF | JTT | CNC | LC |
| -------  | ----- | ----| ----| ----|----|
| **CivilComments** Worst Acc. | 58.6 (1.7) | 58.3 (0.5)|  69.3 (-)| 68.9 (2.1) |70.3 (1.2)|


>**Q2** The authors should show the performance of all methods on all datasets. The authors should report standard deviation over multiple runs.

**A2** Revised Table 1 and Table 2 in the paper with all baseline methods and the standard deviation from 5 seeds are shown below.

| Methods   | Group Info | C-MNIST 0.5 | C-MNIST 1.0 | C-MNIST 2.0 | C-MNIST 5.0 |
|-----------|------------|:-----------:|:-----------:|:-----------:|:-----------:|
| Group Dro |     yes    | 63.12       | 68.78       | 76.30       | 84.20       |
| ERM       |     No     | 35.19(3.49) | 52.09(2.88) | 65.86(3.59) | 82.17(0.74) |
| JTT       |     No     | 53.03(3.89) | 62.9(3.01)  | 74.23(3.21) | 84.03(1.1)  |
| LfF       |     No     | 52.50(2.43) | 61.89(4.97) | 71.03(2.44) | 84.79(1.09) |
| DFA       |     No     | 65.22(4.41) | 81.73(2.34) | 84.79(0.95) | 89.66(1.09）  |
| LC        |     No     | 71.25(3.17) | 82.25(2.11) | 86.21(1.02) | 91.16(0.97) |

| Methods   | Group Info | C-CIFAR 0.5 | C-CIFAR 1.0 | C-CIFAR 2.0 | C-CIFAR 5.0 |
|-----------|------------|:-----------:|:-----------:|:-----------:|:-----------:|
| Group Dro |     yes    | 33.44       | 38.30       | 45.81       | 57.32       |
| ERM       |     No     | 23.08(1.25) | 25.82(0.33) | 30.06(0.71) | 39.42(0.64) |
| JTT       |     No     | 24.73(0.6)  | 26.9(0.3)   | 33.4(1.06)  | 42.2(0.31)  |
| LfF       |     No     | 28.57(1.3)  | 33.07(0.77) | 39.91(0.3)  | 50.27(1.56) |
| DFA       |     No     | 29.95(0.71) | 36.49(1.79) | 41.78(2.29) | 51.13(1.28) |
| LC        |     No     | 34.56(0.69) | 37.34(1.26) | 47.81(2.00) | 54.55(1.26) |

| Methods   | Group Info | Waterbirds  | CelebA      | bFFHQ       |
|-----------|------------|:-----------:|:-----------:|:-----------:|
| ERM       |     No     | 62.85(0.30) | 46.92 (2.17) |56.67 (2.66)|
| JTT       |     No     | 83.8 (1.2)  | 81.5(1.7)    | 65.31(2.5) |
| LfF       |     No     | 78.0 (0.9)  | 77.2(-)      | 62.24 (1.6)|
| DFA       |     No     | 87.72(0.21) | 84.12(1.21) |  82.77(1.40)|
| CNC       |     No     | 88.5 (0.3)  | 88.8 (0.9)  | 68.9 (2.1)  |
| LC        |     No     | 90.5(1.1) |  88.1 (0.8)   | 69.97(1.43) |

####################### Update #######################

There was a typo in the performance of DFA  for bFFHQ in the table, it should be 63.87 (0.31), not 82.77 (1.40). Apologize for the mistake, we have also double-checked other data, and they are correct.

---

### Decision · Program_Chairs · 2023-01-20

**Decision:**

Accept: poster

**Justification For Why Not Higher Score:**

The paper clarity needs significant improvement however given that all reviewers accepted the paper (borderline), I'll advocate for an accept.

**Justification For Why Not Lower Score:**

All reviewers recommend accepting the work; I think it is an important problem studied in the paper

**Metareview: Summary, Strengths And Weaknesses:**

The paper studies the problem of spurious correlation. In particular, authors propose Logit Correction (LC) loss to mitigate the effects of spurious correlations. Further, to synthesize more samples from the minority groups, the authors propose Group MixUp. Reviewers generally liked the problem studied in the paper but clarity of some parts of the paper needs to be improved significantly. AUthors did a good job in the rebuttal to address some other concerns by the reviewers.

**Note From Pc:**

if the above contains the word "oral" or "spotlight" please see: "oral" presentation means -> notable-top-5% and "spotlight" means -> notable-top-25%. As stated in our emails, we are disassociating presentation type from AC recommendations